# Distribution Characteristics of Volatile Organic Compounds and Contribution to Ozone Formation in a Coking Wastewater Treatment Plant

**DOI:** 10.3390/ijerph17020553

**Published:** 2020-01-15

**Authors:** Yuxiu Zhang, Tingting Zang, Bo Yan, Chaohai Wei

**Affiliations:** 1State Key Laboratory of Organic Geochemistry, Guangzhou Institute of Geochemistry, Chinese Academy of Sciences, Guangzhou 510640, China; zhjzhyx@163.com (Y.Z.); zangtingting139@163.com (T.Z.); 2University of Chinese Academy of Sciences, Beijing 100049, China; 3The Environmental Research Institute, South China Normal University, Guangzhou 510031, China; bo.yan@m.scnu.edu.cn; 4The Key Lab of Pollution Control and Ecosystem Restoration in Industry Clusters, Ministry of Education, School of Environment and Energy, South China University of Technology, Guangzhou 510006, China

**Keywords:** volatile organic compounds, ozone formation potential, maximum incremental reactivity (MIR), coking wastewater, anaerobic-oxic-oxic

## Abstract

Ozone pollution, which can be caused by photochemical reactions, has become a serious problem. The ozone formation potential (OFP) is used to describe the photochemical reactivity. Volatile organic compounds (VOCs) are main precursors of ozone formation, and wastewater treatment plants (WWTPs) are important sources of VOCs. Therefore, it is necessary to study the concentration level and OFP of VOCs from WWTPs. In this work, a coking WWTP with anaerobic-oxic-oxic (A/O/O) processes in Shaoguan city, Guangdong province, China, was selected to investigate the characteristics of VOCs at wastewater treatment areas and office areas. The OFP of VOCs was estimated by the maximum incremental reactivity (MIR) coefficient method. Results showed that 17 VOCs were detected, and the total concentration of VOCs was the highest at the raw water tank (857.86 μg m^−3^). The benzene series accounted for 69.0%–86.9% and was the main component of VOCs in the WWTP. Based on OFP data, the top six VOCs contributing most to the OFP were m-xylene, toluene, p-xylene, o-xylene, styrene, and benzene. This study provides field data and information on the environmental risk of VOCs for coking companies and environmental departments. We found that the priority control sources of VOCs were wastewater treatment units because of their larger OFP contributions.

## 1. Introduction

Volatile organic compounds (VOCs) are a class of organic compounds with a saturated vapor pressure greater than 133.32 Pa at room temperature, or a boiling point between 50 and 260 °C at ambient atmospheric pressure, or any volatile organic solid or liquid at normal temperature and pressure [1,2]. Due to low boiling points and high vapor pressures, VOCs can disperse into air from liquid phase at ambient temperature. VOCs are of interest because they are involved in the formation of tropospheric ozone [3,4], thus causing environmental issues such as global greenhouse effect, photochemical smog, and the depletion of stratospheric ozone [5,6]. Long exposure to VOCs may have adverse health risks [7,8,9]. For example, benzene, toluene, and xylenes (BTX) were seen as the most harmful VOCs in petrochemical industry [10], and were classified as priority pollutants by the International Agency for Research on Cancer (IARC) and the United States Environmental Protection Agency (USEPA), causing cancers in lung and kidney [11,12,13]. The USEPA have stated that 34 of 114 kinds of priority organic pollutants in water are volatile organic compounds. Wastewater, especially industrial wastewater, with a large number of VOCs, is the source of VOC emissions during treatment processes [14,15]. More attention has been paid to VOC emissions in the WWTP, especially for unorganized emission sources [16,17]. Therefore, a better understanding of the distribution characteristics of VOCs and estimating the ozone formation potential is important for implementing control strategies.

China is one of the world’s leaders in terms of coal storage and coke production. In 2018, China’s coke production reached 438.2 million tons, accounting for approximately 60% of the world’s total production, and approximately 3.12 × 10^8^ m^3^ of coking wastewater is produced each year [18]. Coking wastewater with complex components contains a large number of organic pollutants, including long-chain alkanes, phenols, polycyclic aromatic compounds, a benzene series and other organic compounds [19,20,21]. These toxic and refractory organic compounds are difficult to completely degrade by biochemical treatment processes and will inevitably escape from wastewater to the atmosphere during the wastewater treatment process. VOCs escape to the atmosphere with aeration and turbulence, causing cancer, teratogenicity and mutation to human health and other direct threats [22]. VOCs cause chemical interactions with oxidizing substances, forming ozone, secondary aerosols and photochemical smog, which have adverse effects on the environment [7,23,24]. The operation site of a coking wastewater treatment station can smell strong and include irritant odor. The total concentration of the benzene series in the gas sample collected above the coking raw water tank is as high as 697.06 μg m^−3^. Coking enterprises are still facing great challenges in meeting the discharge standard and the unorganized discharge of VOCs.

The mechanism of ozone formation has been extensively researched and is well understood. Chemical mechanism modelling is frequently used to estimate the ozone formation potential (OFP). Chemical mechanisms, such as the second generation Regional Acid Deposition Model (RADM2), the Statewide Air Pollution Research Center 07 (SAPRC07), and Carbon Bond 05 (CB05), have been widely used in chemical transport models (CTMs) [25,26,27,28,29,30,31]. These models can accurately estimate OFP but require heavy computation. Maximum incremental reactivity (MIR) is a rapid method for OFP estimation involves multiplying the concentration of each VOC and index. MIR is derived from chamber experiments and thus confined to specific atmosphere conditions [32], and so it is more suitable for OFP estimation.

A better understanding of VOC distribution characteristics is important for discussion on the behavior and fate of VOCs. The objectives of this work were to determine atmospheric VOC concentrations, investigate the distribution characteristics, and estimate the ozone formation potentials of VOCs at different designated sites on and around the coking WWTP. The coking WWTP for this study was located in the Shaoguan coking plant of Guangdong province, China, which adopted an anaerobic-oxic-oxic (A/O/O) biological treatment process that has run stably for approximately 15 years. The wastewater treatment capacity of this coking WWTP is approximately 1500 m^3^ d^−1^, and the annual treatment amount of coking wastewater is approximately 547.500 tons based on a wastewater output of 0.5–0.6 m^3^ when 1 ton of coke is produced. The chemical oxygen demand (COD) of raw coking wastewater is up to 3000–10000 mg L^−1^, containing 558 species of organic matters [20]. Organic compounds will partly volatilize during the treatment process under the action of aeration and turbulence. Therefore, the emission and distribution characteristics of VOCs in this WWTP need to be investigated. This study provided supporting data to understand ozone generation, secondary aerosol, and particulate matter of 2.5 μm and smaller in size (PM 2.5) hazards caused by VOCs in further research.

## 2. Materials and Methods

### 2.1. Location of the Coking WWTP

The coking WWTP investigated for this study is located in the Shaoguan coking plant in Guangdong province, China. This WWTP has been running stably for approximately 15 years. The coking wastewater treatment capacity of this WWTP is approximately 1500 m^3^ d^−1^. The A/O/O biological treatment process was applied. Indicators of coking wastewater such as temperature, pH, the COD and dissolved oxygen (DO) at various stages are shown in Table 1. Raw coking wastewater with high COD contains a large amount of organic matter—part of which already consists of VOCs, and a further part may be degraded into VOCs during the treatment process.

### 2.2. VOC Sampling

To obtain the concentrations of VOCs, air sampling was conducted over the main wastewater treatment areas and in the office areas based on the USEPA TO-17 method [33,34]. The air sampling locations in the WWTP are illustrated in Figure 1. At the wastewater treatment area, sampling sites include the raw water tank (RWT), the ammonia stripping tower (AST), the anaerobic tank (A), the anterior aerobic tank (O1), the posterior aerobic tank (O2), and the effluent tank (ET). At the office area, sampling sites include office rooms and a main aisle. Blank samples were also collected at the same time. Four parallel samples were collected at each sampling site. Samples were collected from August to October 2016.

The gas-collecting hood above the wastewater was used to collect gaseous VOCs emitted from the tanks. One end of the gas-collecting hood was connected to a pumping air sampler (TH-150 Automatic Flow Control Medium Volume TSP Sampler, Wuhan Tianhong Instruments Co., Ltd., China) and a pre-treatment Tenax-TA tube (a glass tube, with an inner diameter of 5 mm and an outer diameter of 6 mm, containing 180 mg of 60–80 mesh Tenax TA adsorbent inside). The Tenax-TA tubes were obtained from Zhengzhou Spectrum Analysis Technology Co., Ltd., China, and pretreated under the conditions of a temperature of 300 °C, a nitrogen flow rate of 50 mL min^−1^, and a duration of 0.5 h. The other end was connected to an activated carbon adsorption tube to prevent water or carbon dioxide in the air from contaminating samples. To ensure all gases in the gas-collecting hood discharge from the tank, the pump was run for approximately 20 min before connecting the Tenax-TA tube. Recommended by the USEPA TO-17 method, a gas flow rate of 200 ± 5 mL min^−1^ and a sampling time of 0.5 h were adopted in the actual sampling process of this study. Both ends of the sampling tube were sealed with stainless steel end fittings and a cap. Samples were refrigerated below 4 °C and analyzed within seven days. To reduce the impact of both background and nearby facilities, all sampling was conducted on days with wind speed < 0.5 m s^−1^).

### 2.3. VOCs Analytical Method

Based on the USEPA TO-17 method [33,34,35], atmospheric samples were analyzed using gas chromatography-mass spectrometry (GC–MS) and thermal desorption equipment (Gerstel Varian, USA: TDS3-3800GC-4000MS) to determine the concentrations of VOCs. The temperature of desorption was 280 °C. VOCs were desorbed from the Tenax-TA by helium for 10 min, cooled and collected by a cooling trap, and then blown into the GC, using a fused silica capillary column (HP-VOC, 60 m × 0.32 mm × 1.80 µm) to separate gases. A temperature program was used: 35 °C for 5 min, ramped to 100 °C at 2 °C min^−1^, and then ramped to 220 °C at 5 °C min^−1^, which was maintained for 5 min (total time approximately 67 min). VOCs were quantified using standard compounds containing 54 VOCs. The standard solution containing 54 VOC compounds, each at a concentration of 200 mg L^−1^, was obtained from Organic Standards Solutions International Company of USA. Methanol as the dilution solvent was purchased from Merck (Darmstadt, Germany). Proper amounts of standard solution were used and diluted to 1.00 mL with methanol to prepare a series of standard stock solutions at different concentrations for calibration.

For quality assurance and control, duplicate analyses were conducted, and blank samples were analyzed. Quantitative analysis was obtained by calibration curves based on peak areas by injecting 10 µL standard samples into sampling tubes at concentrations of 0.25, 0.50, 1.00, 2.50, 5.00, 10.00, 40.00 µg mL^−1^, respectively. The calibration curve was developed for each individual VOC, and the correlation coefficient of each VOC calibration curve >0.99. Method detection limits for each VOC were 0.077–0.48 ng mL^−1^.

### 2.4. Calculation of Ozone Ormation Potential

Many species of VOCs play a key role in the formation of photochemical smog and ozone formation. The ozone formation potential (OFP) of a VOC depends upon the concentration of the VOC and also the reactivity in ambient air. The ozone formation potential is generally used to evaluate the photochemical activity of VOCs in ambient air [36,37]. The OFP of VOCs depends on the concentration and the reactivity of the species in the atmosphere. Different methods can be used to define the chemical reaction activity of VOCs [38,39]. There is a maximum incremental reactivity (MIR) method and an equivalence method [40,41,42]. In this study, the contribution of VOCs to the activity of the chemical reaction was evaluated by the ozone potential method. The ozone generation potential of VOCs discharged unorganized from coking WWTPs is estimated by the MIR coefficient method, and the calculation formula is as follows.
(1)$${\mathrm{OFP}}_{i}=C_{i}\times {\mathrm{MIR}}_{i}$$
where OFP_i_ is the ozone formation potential of a VOC, μg m^−3^; *C_i_* is the actual measured mass concentration of a VOC; MIR_i_ is the maximum incremental reactivity of a VOC [43,44].

## 3. Results and Discussion

### 3.1. Chemical Composition of Atmospheric VOCs

Understanding the chemical composition and the concentration of VOCs in the air phase at different stages of coking WWTPs will help to identify the distribution characteristics of VOCs. In total, 17 VOCs were detected, and the lack of alkanes, alkenes, and alkynes was due to the deficiency of the sampling method. Therefore, the total VOCs (TVOC) was defined as the sum of the detected 17 VOCs, including 12 species of a benzene series (benzene, toluene, o-xylene, p-xylene, m-xylene, styrene, 1,3,5-trimethylbenzene, 1,2,4-trimethylbenzene, n-propylbenzene, isopropylbenzene, n-butylbenzene, and p-isopropyltoluene), three chlorinated hydrocarbons (chloroform, carbon tetrachloride, and trichloroethylene), and two chlorinated benzene compounds (chlorobenzene and 2-chlorotoluene). Figure 2 shows the concentrations of VOCs and TVOC at various treatment stages of the coking WWTP. The total amount of VOC emissions from each treatment process varied from 28.56 ± 3.96 µg m^−3^ to 857.86 ± 131.30 µg m^−3^, with an average concentration of 266.87 µg m^−3^. The highest TVOC concentration was observed in the RWT at 857.86 ± 131.30 µg m^−3^, followed by A, AST, O1, O2, and ET units at 28.56 ± 3.96 µg m^−3^ to 533.87 ± 63.14 µg m^−3^. The benzene series displayed the maximum emission at 697.06 ± 105.95 µg m^−3^, followed by chlorinated hydrocarbons at 84.45 ± 11.21 µg m^−3^, and chlorinated benzene compounds at 76.35±14.14 µg m^−3^. The concentration of VOCs around office areas varied from 45.66 to 72.02 µg m^−3^. Based on the health toxicity values from the United States Environmental Protection Agency Integrated Risk Information System (USEPA-IRIS), the acute reference concentration (R_f_C) of benzene is 29 µg m^−3^, and the chronic R_f_C of benzene is 30 µg m^−3^. VOC emissions from treatment stages that have spread to the office area may pose environmental and health risks.

At all the treatment stages, benzene, toluene, xylenes, styrene, chlorobenzene, and chloroform were ubiquitous and were the main pollutants, with a total concentration of 27.52–830.89 µg m^−3^, accounting for > 96% of the total concentration of 17 VOCs. Benzene has the highest concentration of 180.49 µg m^−3^ (accounting for approximately 21.0%), followed by styrene (15.9%) and toluene (13.5%) at the RWT stage (Figure 3). Benzene, toluene, xylenes, styrene, chlorobenzene, and chloroform were the hazardous gaseous materials.

### 3.2. Distribution Characteristics of Atmospheric VOCs

The concentrations of VOCs in different treatment stages followed this order: raw water tank > anaerobic tank > ammonia stripping tower > anterior aerobic tank > posterior aerobic tank > effluent tank (Figure 4). Results show that the concentrations of VOCs emitted from the raw wastewater tank were relatively high, especially for the benzene series. The raw wastewater tank was a main source of VOCs, because the VOC concentration in wastewater is one of the important influencing factors of VOC emissions, as shown in our previous work [45].

Apart from the composition of the wastewater, factors such as facility design, ventilation, aeration, water retention time, and sludge retention time, among others, contribute to the reduction of VOCs. The concentrations of gaseous VOCs are related to influencing factors such as ventilation, chemical processes, physicochemical properties, etc. Physicochemical properties reflecting VOC volatility are boiling point, water solubility, the Henry coefficient (H*_c_*), and vapor pressure. There is a certain correlation between gaseous VOC concentrations and boiling point, molecular weight, water solubility, H*_c_*, the log of the octanol–water partition coefficient, and steam pressure [45]. Considering the same chemical processes and conditions for the xylenes, and the concentration of o-xylene in wastewater was higher than that of m-xylene or p-xylene, o-xylene had the lowest gaseous concentration because of lower volatility and higher solubility [46]. Excessive aeration sped up VOC emission from open tanks [47]. However, insufficient aeration can also result in higher VOC emission [48]. Thus, a well-operated WWTP can better reduce the generation of VOCs than lacking good operating conditions [49].

### 3.3. Cluster Analysis

Cluster analysis is an important technique in data mining and exploratory data analysis. Based on VOC concentrations at each treatment unit, cluster analysis was conducted. The pseudo-F statistic was called Calinski and Harabasz-F index and calculated to obtain the optimal classification number of clusters by the cluster stop command using Stata software which is statistical software for data science from StataCorp, Texas, USA (Table 2). A higher pseudo-F statistic indicates an optimal classification number. Based on the cluster stop rule, VOCs were divided into four categories. From Figure 5, the 17 VOCs detected can be divided into four categories: (1) benzene (number 1); (2) toluene and chloroform (number 2 and 13); (3) m-xylene and styrene (number 3 and 12); and (4) other categories. Benzene, with the largest gaseous concentration, forms a class by itself, maybe because the concentration of benzene in the wastewater phase was the largest. Toluene and chloroform are classified maybe because removal efficiencies at RWT, AST, and A units are low, and the concentrations do not change greatly. Styrene and m-xylene are classified maybe because removal efficiencies at AST unit are higher than at RWT unit, and the concentration changes greatly at AST unit. Other categories are classified due to their concentrations at low levels. It can be seen from the previous work that VOC emissions from wastewater into the air are positively correlated with VOC concentrations in the wastewater, and physicochemical properties and treatment technologies are important factors in VOC removal [45].

### 3.4. The OFP of VOCs in a Coking WWTP

Based on the concentration of VOCs and the maximum incremental reactivity (MIR), the effect of VOCs on ozone formation was estimated. The ozone formation potential (OFP) of wastewater treatment area and office area is calculated according to Equation 1 (Table 3). Results show that the total OFP in the raw water tank is the highest (2960.49 μg m^−3^), followed by in the ammonia stripping tower (1554.51 μg m^−3^) and in the anaerobic tank (1546.68 μg m^−3^), and that in the management area was relatively low. The total OFP at different treatment stages was in the range of 81.49–2960.49 μg m^−3^, and the top 10 greatest contributions to the total OFP in the coking WWTP were m-xylene, toluene, p-xylene, o-xylene, styrene, benzene, 1,2,4-trimethylbenzene, 1,3,5-trimethylbenzene, chlorobenzene, and isopropylbenzene (Figure 6). The OFP of benzene, toluene and xylene (BTX) account for 85.6%, and these are the main contributors to ozone generation. Previous studies have also shown that BTX is the main contributor to ozone generation [50,51]. In this paper, the value of the ozone formation potential is higher than the recommended air quality limit (100 μg m^−3^) [52]. A high OFP value increases ozone generation and causes adverse health effects [53,54,55], which means that BTX compounds with a high MIR value show photochemical activity [42,56].

## 4. Conclusions

VOCs, as harmful that chemicals affect ozone formation, modify the atmospheric chemistry by participating in various photochemical reactions and generate other secondary pollutants. In total, 17 kinds of VOCs were detected in the air samples, including a benzene series, chlorinated hydrocarbons, and chlorobenzenes emitted from a coking WWTP, with a total concentration range of 28.56–857.86 μg m^−3^ at the wastewater treatment area, and 45.66–72.02 μg m^−3^ at the office area. The main sources of VOCs were the raw water tank (857.86 μg m^−3^), followed by the anaerobic tank (533.87 μg m^−3^), the ammonia stripping tower (521.68 μg m^−3^), the anterior aerobic tank (190.08 μg m^−3^), the posterior aerobic tank (52.87 μg m^−3^), and the final effluent (28.56 μg m^−3^). This study reports the levels of VOCs during the wastewater treatment process. The benzene series accounted for 69.0%–86.9% and were the main component of VOCs in the WWTP. Based on OFP data, the total OFP was 81.49–2960.49 µg m^−3^ at the wastewater treatment area, and the average total OFP levels (1136.27 ± 154.11 μg m^−3^) in this study were higher than the air quality guidelines proposed by the World Health Organization (WHO) at 120 μg m^−3^. The benzene series was the main contributor (>99.0%) to ozone formation and was harmful to human health and air pollution. The top six VOCs contributing most to the OFP were m-xylene (36.0%), toluene (20.7%), p-xylene (13.6%), o-xylene (10.7%), styrene (6.9%), and benzene (5.3%). The total VOC concentrations from this coking WWTP in this study were one order of magnitude higher than that of a municipal WWTP [57]. The OFPs in each unit of this coking WWTP (81.49–2960.49μg m^−3^) had similar levels of each unit of a sewage sludge composting plant at 57.5–3582.5 μg m^−3^ [58].

Such high levels of VOCs and OFPs obtained in this study highlight the need to control the levels of these pollutants in ambient air in the coking WWTP to mitigate the associated photochemical activity and air pollution. VOC emissions from coking WWTPs should not be ignored and should be controlled. Based on the findings of the present study, we make more suggestions such as reducing the VOC concentration in wastewater by improving treatment efficiency, reducing VOC emissions by adsorption, catalytic oxidation, or covering wastewater tanks. These suggestions will support the transition from poor to healthier air in coking WWTPs by reducing the levels of these pollutants in ambient air.

## Figures and Tables

**Figure 1 ijerph-17-00553-f001:**
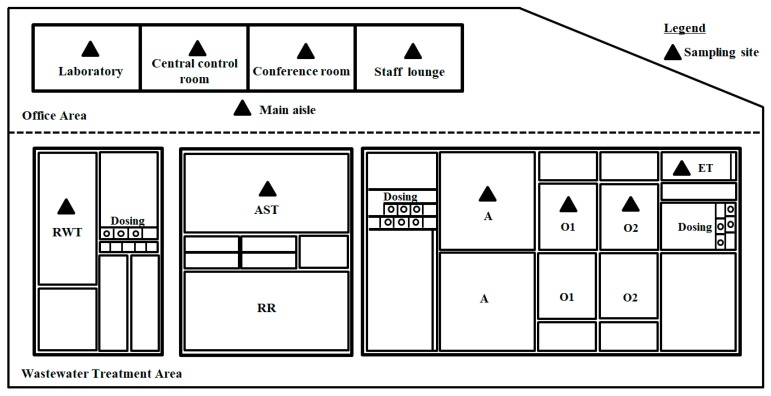
A/O/O treatment processes and sampling sites of the coking WWTP. Wastewater treatment areas include the raw water tank, the ammonia stripping tower, the anaerobic tank, the anterior aerobic tank, the posterior aerobic tank, and the effluent tank. A/O/O = anaerobic-oxic-oxic; WWTP = wastewater treatment plant; RWT = raw water tank; AST = ammonia stripping tower; A = anaerobic tank; O1 = anterior aerobic tank; O2 = posterior aerobic tank; ET = effluent tank.

**Figure 2 ijerph-17-00553-f002:**
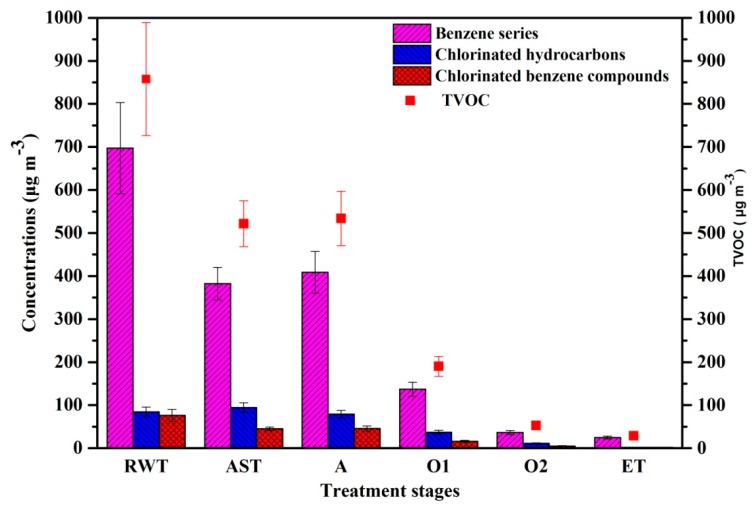
Concentrations of VOCs and TVOC at various treatment stages of the coking WWTP. TVOC represents the total concentrations of all detected VOCs. The treatment stages include the raw water tank, the ammonia stripping tower, the anaerobic tank, the anterior aerobic tank, the posterior aerobic tank, and the effluent tank. VOCs = volatile organic compounds; TVOC = total VOCs; WWTP = wastewater treatment plant; RWT = raw water tank; AST = ammonia stripping tower; A = anaerobic tank; O1 = anterior aerobic tank; O2 = posterior aerobic tank; ET = effluent tank.

**Figure 3 ijerph-17-00553-f003:**
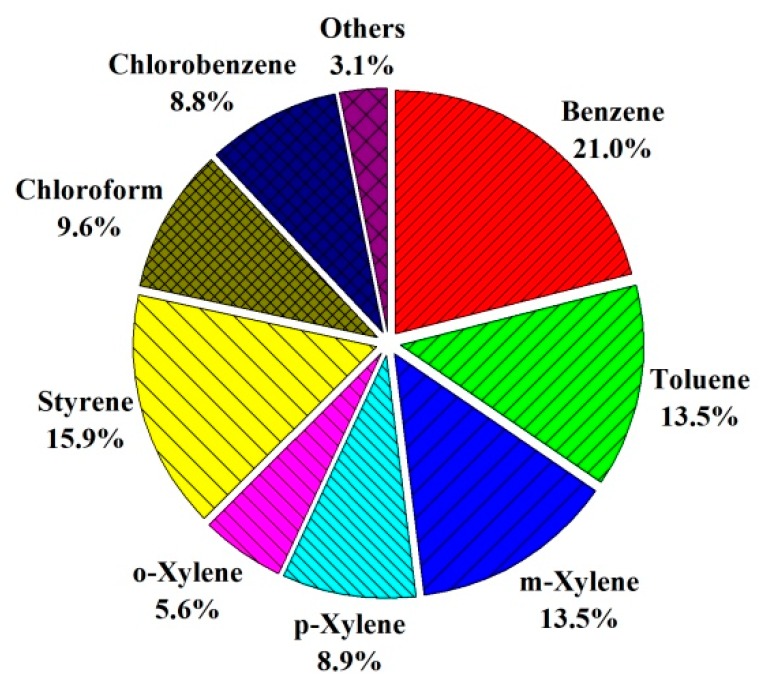
Main hazardous gaseous VOCs above the raw wastewater tank.

**Figure 4 ijerph-17-00553-f004:**
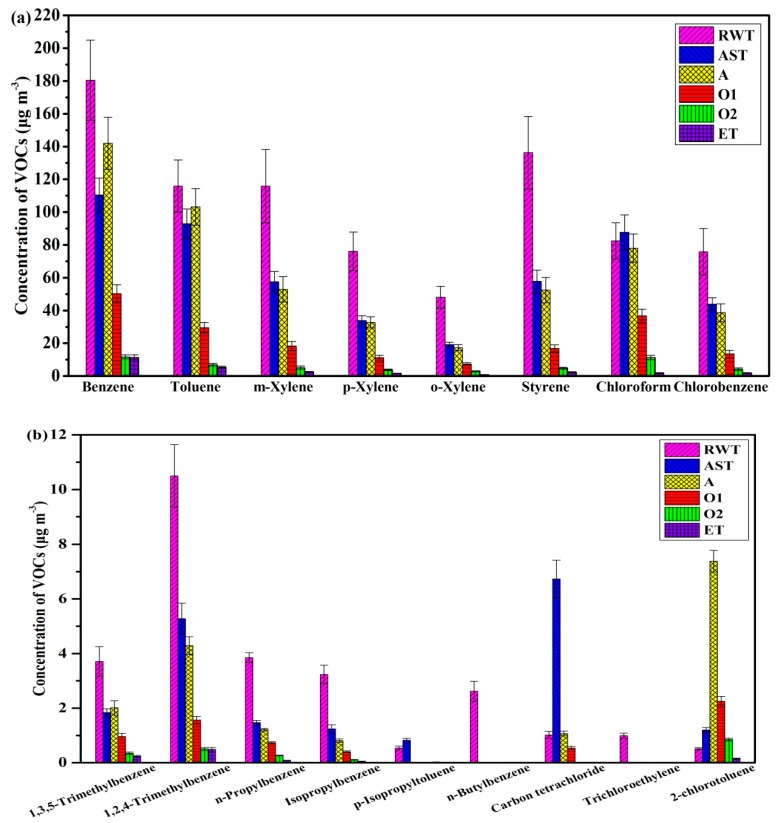
Concentrations of VOCs in the gas phase at different stages of the coking WWTP (**a**) VOCs with high concentration, (**b**) VOCs with low concentration. The treatment stages include the raw water tank, the ammonia stripping tower, the anaerobic tank, the anterior aerobic tank, the posterior aerobic tank, and the effluent tank. RWT = raw water tank; AST = ammonia stripping tower; A = anaerobic tank; O1 = anterior aerobic tank; O2 = posterior aerobic tank; ET = effluent tank.

**Figure 5 ijerph-17-00553-f005:**
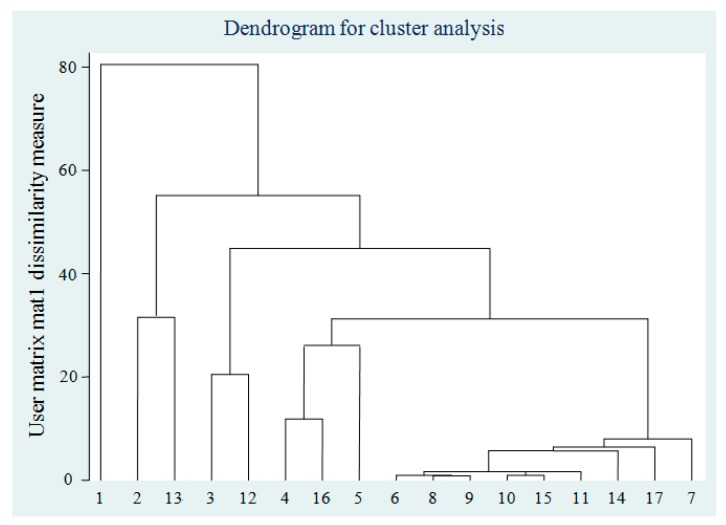
Dendrogram for cluster analysis of detected VOCs. The numbers 1 to 17 on the X-axis represent the following VOC species in order: benzene, toluene, m-xylene, p-xylene, o-xylene, 1,3,5-trimethylbenzene, 1,2,4-trimethylbenzene, n-propylbenzene, isopropylbenzene, p-isopropyltoluene, n-butylbenzene, styrene, chloroform, carbon tetrachloride, trichloroethylene, chlorobenzene, and 2-chlorotoluene.

**Figure 6 ijerph-17-00553-f006:**
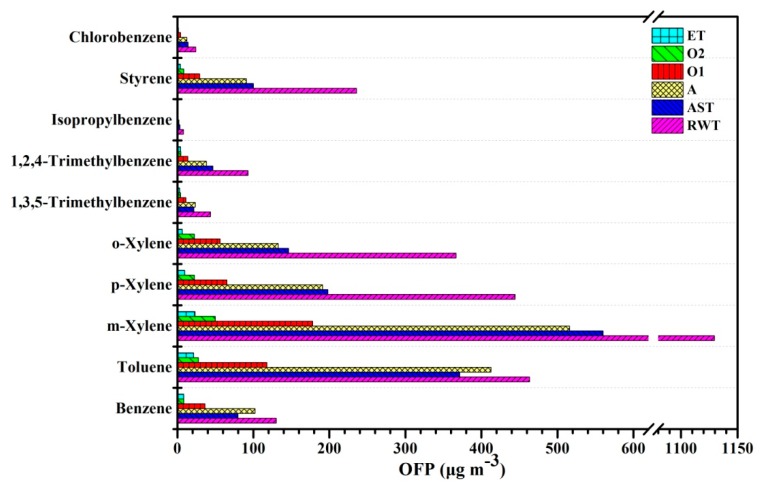
The top 10 greatest contributions to the total OFP in the coking WWTP. RWT = raw water tank; AST = ammonia stripping tower; A = anaerobic tank; O1 = anterior aerobic tank; O2 = posterior aerobic tank; ET = effluent tank.

**Table 1 ijerph-17-00553-t001:** Indicators of the coking wastewater at various stages.

Indicators	Temperature (°C)	pH	COD (mg L^−1^)	BOD_5_ (mg L^−1^)	DO (mg L^−1^)
RWT	35 ± 5	10 ± 0.5	3300 ± 600	1500 ± 50	-
AST	30 ± 3	9.5 ± 0.4	2700 ± 300	1350 ± 50	-
A	28 ± 2	7.6 ± 0.3	850 ± 100	350 ± 30	0.1–0.3
O1	27 ± 2	7.3 ± 0.3	300 ± 70	80 ± 10	1.5–2.0
O2	27 ± 2	7.6 ± 0.3	200 ± 50	20 ± 5	2.5–3.0
ET	23 ± 2	7.5 ± 0.4	80 ± 15	10 ± 3	-

Note: COD = chemical oxygen demand; BOD_5_ = biochemical oxygen demand (5 day); DO = dissolved oxygen; RWT = raw water tank; AST = ammonia stripping tower; A = anaerobic tank; O1 = anterior aerobic tank; O2 = posterior aerobic tank; ET = effluent tank.

**Table 2 ijerph-17-00553-t002:** The optimal classification number of clusters.

Number of Clusters	Calinski and Harabasz Pseudo-F Index
3	44.07
4	61.48
5	19.04

**Table 3 ijerph-17-00553-t003:** The MIR values and OFPs of VOCs in a coking wastewater treatment plant.

Number	VOCs	MIR	RWT	AST	A	O1	O2	ET	Office Room	Aisle
1	Benzene	0.72	129.95	79.53	102.23	36.17	8.40	8.22	4.41	14.05
2	Toluene	4.00	463.52	371.36	412.60	117.88	27.68	21.36	36.20	59.52
3	*m*-Xylene	9.75	1129.64	560.14	515.87	178.13	50.02	23.21	69.71	83.17
4	*p*-Xylene	5.84	444.42	198.03	191.03	65.12	22.25	9.58	21.26	30.13
5	*o*-Xylene	7.64	366.72	146.38	132.48	56.15	22.31	6.42	15.89	20.78
6	1,3,5-Trimethylbenzene	11.76	43.63	21.64	23.64	11.29	4.12	2.82	1.76	8.70
7	1,2,4-Trimethylbenzene	8.87	93.14	46.83	38.05	13.84	4.52	4.35	11.26	14.81
8	n-Propylbenzene	2.03	7.82	2.98	2.46	1.48	0.55	0.16	0.63	0.67
9	Isopropylbenzene	2.52	8.14	3.12	2.04	1.01	0.28	0.13	0.45	0.00
10	p-Isopropyltoluene	6.23	3.36	5.11	0.00	0.00	0.00	0.19	3.99	0.50
11	n-Butylbenzene	2.36	6.18	0.00	0.00	0.00	0.00	0.00	3.07	0.45
12	Styrene	1.73	235.76	99.89	90.69	29.19	8.30	4.03	9.31	13.01
13	Chloroform	0.022	1.81	1.93	1.72	0.81	0.25	0.04	0.09	0.11
14	Carbon tetrachloride	0	0.00	0.00	0.00	0.00	0.00	0.00	0.00	0.00
15	Trichloroethylene	0.64	0.63	0.00	0.00	0.00	0.00	0.00	0.00	0.00
16	Chlorobenzene	0.32	24.27	14.05	12.34	4.33	1.35	0.56	1.34	1.64
17	2-Chlorotoluene	2.92	1.49	3.50	21.55	6.57	2.48	0.44	0.73	1.52
-	Total OFP	-	2960.49	1554.51	1546.68	521.96	152.50	81.49	180.11	249.06

Note: MIR = maximum incremental activity; RWT = raw water tank; AST = ammonia stripping tower; A = anaerobic tank; O1 = anterior aerobic tank; O2 = posterior aerobic tank; ET = effluent tank. The unit of columns 3–10 is μg m^−3^.

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
