# Peer review of "Distribution Characteristics of Volatile Organic Compounds and Contribution to Ozone Formation in a Coking Wastewater Treatment Plant"

_ijerph, 2020, doi:10.3390/ijerph17020553_

Round 1

Reviewer 1 Report

Zhang et al. reported an investigation using gas chromatography-mass spectrometry on VOCs sampled from different locations within a wastewater treatment plant located in Shaoguan, China.

Overall, the English language is good. The manuscript is of interest and it well fits with the aims and scope of the journal. However, some aspects of the paper need to be addressed before I can recommend this work for publication.

Abstract: Please include what A/O/O stands for.

Line 109: pumping air sampler – Details? Brand?

Line 110: Tenax TA tubes – Where these commercially obtained (Brand?) Pre-packed?

Line 112 & 113: “The gas flow rate was (200-300) mL min-1, and the sampling time was (0.5-1) h.” The control of flow rate and sampling period is crucial for the accurate quantification of VOCs. Therefore, I do not understand why the authors have mentioned a range of flows (200-300) mL min-1 and sampling periods (0.5-1) h. Please make yourself clear as regards to the actual measurement conditions, were these not always constant?

Line 125: “temperature of the thermal desorption equipment was 280 °C”. Temperature of desorption should be given instead. “…were blown off” should say – the VOCs were desorbed…

Line 129: “VOCs were quantified using standard compounds containing 54 VOCs, each at a concentration of 200 mg mL-1.” Where these purchased from Merck ?

Line 130: “Dichloromethane, acetone, hexane and methanol were purchased from Merck.” How were these used? Details of solution preparation should be given.

Line 169: “VOCs emission from treatment stages spread to the office area may pose environmental and health risks.” You need to be more specific, the workplace exposure limits for the detected compounds should be given.

Figure 2: Why is the VOC concentration in the RWT the main source of VOCs?

Line 191: “Among the xylenes, o-xylene had the lowest concentration due to its low volatility”

I do not think the difference in concentrations is related to low volatility of o-xylene since isomers have a fairly similar boiling point, as expected. This can certainly be accounted by the chemical processes involved and other factors such as ventilation, considering the temperature is constant. This should be discussed in the manuscript in detail.

m-xylene, bp = 412 K

p-xylene, bp = 411 K

o-xylene, bp = 417 K

(boiling points (bp) given by NIST)

Figure 4: X-axis the numbers should be replaced by the name of compound, if possible. It is rather difficult to analyse the figure.

Reviewer 2 Report

This paper reported VOC measurement result at coking wastewater treatment plant in China. It would be useful information.

[detailed comments]

line16:   Is this explanation correct?

Do you mean

"Ozone pollution, which can be caused by photochemical reaction , has become a serious problem" ?

Line 20:  "A/O/O" appears first time. Please add here "anaerobic-oxic-oxic".

Line 39  "the depletion of stratospheric ozone"  is only for chlorinated VOCs and explanation here is about ozone formation.  It is better to move the explanation about influence to ozone layer after “trophospheric ozone formation [5,6]”

2.2 VOC sampling:  How many samples were taken for one location? In Figure 2 etc. error bar are shown in bar graph.

Line 110-111:  What is “influence of air “ ?

line 200:  I do not understand why you explained “3.3 cluster analysis” in this paper.

Reviewer 3 Report

The aim of the present study is to determine atmospheric VOCs  concentrations, investigate the distribution characteristics, and estimate ozone formation potentials of VOCs at different designated sites on and around the coking WWTP. The topic is relevant to this journal, but I would suggest major revision before the current version could be considered for publication.

Major revision

3. Results and discussion

Page 4 line 164

Authors write “The highest TVOC concentration was observed in the RWT at 857.86±131.30 μg m-3, followed by AST, A, 164 O1, O2, ET units at 28.56±3.96 μg m-3 to 533.87±63.14 μg m-3, but figure 2 shows that the concentration of TVOCs is higher in A than in  AST.

Figure 4

The authors should make figure 4 more understandable by inserting for example an arrow to explain the enlargement of the graph from point 6 to 17.

Cluster analysis

Authors should better explain cluster analysis: what is the pseudo -F index, how is it obtained, what do high or low values of this index indicate?

The authors should also explain the dendogram: what is the meaning? in this case what does it mean?

Page 7, line 219.

Results show that the total OFP  in the raw water tank is the highest (2927.17 μg m-3), followed by ammonia stripping tower (1531.39 219 μg m-3) and anaerobic tank (1531.73 μg m-3), and the management area was relatively low.

Results reported indicate that  the total OFP  in the raw water tank is the highest (2927.17 μg m-3), followed by anaerobic tank (1531.73 μg m-3) and not from ammonia stripping tower (1531.39 219 μg m-3).

Conclusions

Conclusions should be extended by discussing the results of this study with those obtained in other studies.

Round 2

Reviewer 1 Report

The authors have answered to all the comments accordingly.

I now recommend this paper for publication.

Reviewer 3 Report

Authors reviewed the manuscript according to my comments,  the manuscript can be accepted in the revised version.